# Cohort profile: the Pregnancy Risk Infant Surveillance and Measurement Alliance (PRISMA) – Pakistan

Sabahat Naz [ORCID],[1] Ali Jaffar,[1] Nida Yazdani,[2] Muhammad Kashif,[1] Zaid Hussain,[1] Uzma Khan,[2] Fouzia Farooq,[3] Muhammad Imran Nisar,[1] Fyezah Jehan [ORCID],[1] Emily Smith,[3] Zahra Hoodbhoy [ORCID] [1]

[1]Department of Pediatrics and Child Health, The Aga Khan University, Karachi, Sindh, Pakistan
[2]VITAL Pakistan, Karachi, Pakistan
[3]Department of Global Health, Milken Institute School of Public Health, The George Washington University, Washington, Columbia, USA

**Correspondence to**
Dr Zahra Hoodbhoy;
zahra.hoodbhoy@aku.edu

## ABSTRACT

**Purpose** Pakistan has disproportionately high maternal and neonatal morbidity and mortality. There is a lack of detailed, population-representative data to provide evidence for risk factors, morbidities and mortality among pregnant women and their newborns. The Pregnancy Risk, Infant Surveillance and Measurement Alliance (PRISMA) is a multicountry open cohort that aims to collect high-dimensional, standardised data across five South Asian and African countries for estimating risk and developing innovative strategies to optimise pregnancy outcomes for mothers and their newborns. This study presents the baseline maternal and neonatal characteristics of the Pakistan site occurring prior to the launch of a multisite, harmonised protocol.

**Participants** PRISMA Pakistan study is being conducted at two periurban field sites in Karachi, Pakistan. These sites have primary healthcare clinics where pregnant women and their newborns are followed during the antenatal, intrapartum and postnatal periods up to 1 year after delivery. All encounters are captured electronically through a custom-built Android application. A total of 3731 pregnant women with a mean age of 26.6±5.8 years at the time of pregnancy with neonatal outcomes between January 2021 and August 2022 serve as a baseline for the PRISMA Pakistan study.

**Findings to date** In this cohort, live births accounted for the majority of pregnancy outcomes (92%, n=3478), followed by miscarriages/abortions (5.5%, n=205) and stillbirths (2.6%, n=98). Twenty-two per cent of women (n=786) delivered at home. One out of every four neonates was low birth weight (<2500 g), and one out of every five was preterm (gestational age <37 weeks). The maternal mortality rate was 172/100 000 pregnancies, the neonatal mortality rate was 52/1000 live births and the stillbirth rate was 27/1000 births. The three most common causes of neonatal deaths obtained through verbal autopsy were perinatal asphyxia (39.6%), preterm births (19.8%) and infections (12.6%).

**Future plans** The PRISMA cohort will provide data-driven insights to prioritise and design interventions to improve maternal and neonatal outcomes in low-resource regions.

**Trial registration number** NCT05904145.

## STRENGTHS AND LIMITATIONS OF THIS STUDY

⇒ The prospective nature of the Pregnancy Risk, Infant Surveillance and Measurement Alliance Pakistan cohort enables the collection of high-quality data from attendees of the primary healthcare (PHC) available in the community within the defined geographical area.

⇒ This is a unique longitudinal cohort of attendees of the PHC available in the community in Pakistan with the ability to leverage an existing demographic surveillance system to capture early pregnancies and follow these women post partum with low attrition rates.

⇒ Although women are encouraged to deliver at the recommended referral facility for obtaining their information, nearly 36% delivered at other health facilities or their homes. In such cases, data on key labour and delivery outcomes are self-reported, thus limiting its interpretation.

## INTRODUCTION

Population-based longitudinal birth cohorts provide important data to study a life course approach to in utero exposures and aetiological factors that contribute to developing diseases early and later in life.[1] The advantages of such cohort studies include standardised data collection, a systematic approach in following participants throughout the study and the generalisability of findings to the larger population.[1] In addition, it allows investigators to collect accurate information on the exposures, outcomes, potential confounders and other important characteristics at the individual as well as population level.[2] Early identification of these risk factors may provide an opportunity to design and implement population-based preventive strategies to optimise individual and population-level outcomes.[3]

Establishing robust perinatal datasets in low-resource settings is essential for tracking local demographic shifts related to risk

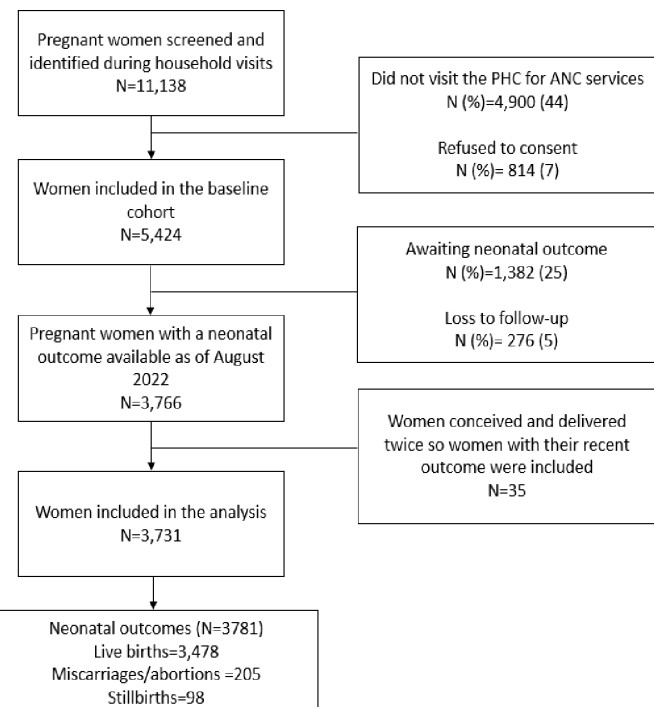

**Figure 1** Flow diagram for PRISMA enrolment status January 2021 to August 2022. ANC, antenatal care; PHC, primary healthcare; PRISMA, Pregnancy Risk, Infant Surveillance and Measurement Alliance.

factors, morbidities and mortality outcomes among pregnant women and their newborns.[4] This information can also aid in the development of stratified care pathways for efficient resource utilisation, thus improving outcomes.[3] The Demographic Health Survey (DHS) and the Maternal Newborn Health Registry of the NICHD Global Network for Women's and Children's Health Research are examples that have generated a wealth of data over a decade in low-income and middle-income countries, including Pakistan.[4 5] However, these datasets are either cross-sectional (i.e., DHS) or have limited timepoints during the pregnancy care continuum. Furthermore, these datasets focus on maternal and neonatal mortality as primary outcomes and lack detailed clinical and laboratory data on major morbidities in the mother and infant.

To address these gaps, the Pregnancy Risk Infant Surveillance and Measurement Alliance (PRISMA) Maternal and Newborn Health study is being conducted across five sites in South Asia (India, Pakistan) and sub-Saharan Africa (Zambia, Kenya, Ghana) to longitudinally assess pregnancy risk factors and their relationships with poor pregnancy outcomes, such as stillbirth, neonatal morbidity and mortality, and maternal mortality and severe maternal and infant morbidity. This multicountry, open cohort study aims to provide a harmonised dataset to quantify the burden of maternal, fetal and neonatal risk factors and understand vulnerabilities, morbidity and mortality.

The objective of this paper was to provide an overview of the baseline characteristics of women in the study catchment area for the Pakistan site before the initiation of the PRISMA study.

## Cohort description
### Study design and participants
In Pakistan, the PRISMA study is being conducted in two periurban community settings in Karachi, namely Rehri Goth and Ibrahim Hyderi; the areas comprise approximately 23 170 households and a population of 227 549 individuals (refer to online supplemental figure 1).

The existing maternal and child health surveillance system at these sites will be leveraged for the early identification of pregnancies for the PRISMA cohort. The Department of Pediatrics and Child Health at the Aga Khan University (AKU) has been conducting maternal and child surveillance at the periurban coastal field sites since 2003.[6] This system gathers data on vital events regarding married women of reproductive age and children under 5 years of age. Records of births, deaths, pregnancies and migration events are documented through quarterly household visits by community health workers (CHWs) using the existing household rosters of these sites.

Pregnant women captured through pregnancy surveillance are encouraged to visit the primary healthcare (PHC) clinics located within the community for antenatal care (ANC). These PHCs are run by AKU and are not part of the public health system. ANC, along with laboratory and ultrasound procedures, is offered free of cost for all pregnant women from the catchment area by community midwives located at these centres. Pregnant women aged 15–49 years with a confirmed intrauterine pregnancy of gestational age (GA)<20 weeks via standard ultrasound (GE Vivid IQ)[7] are eligible to participate in the study. A total of 3731 pregnant women with a neonatal outcome of 3781 between January 2021 and August 2022 serve as a baseline for PRISMA Pakistan. The flow chart of this population is described in figure 1.

### Measures and procedure
At baseline, pregnant women visit the PHC located within the study catchment area for ANC services. At the first ANC visit, the pregnancy history of the women along with anthropometry, blood pressure and other vital information regarding the current pregnancy is obtained. The pregnant woman has her ultrasound to establish the GA at the first visit followed by laboratory investigations (including haemoglobin, hepatitis B and C screening). The subsequent ANC visit dates are given as per WHO recommendations for ANC visits.[8] During delivery, women are provided transport to our partnering hospital to ensure skilled labour and delivery. In cases where women give birth at home or at another facility besides our partnering facility, CHWs visit the household to obtain information regarding labour and delivery, complications, if any, and perform newborn measurements (weight, length and head circumference) within 72 hours of life. In case of a maternal or neonatal mortality, a

**Table 1** Sociodemographic and clinical characteristics of women included and excluded in the baseline cohort of PRISMA Pakistan

| Characteristics | Women included in the analysis (n=3731) mean±SD, median (range) or n (%) | Women who did not receive ANC services or did not consent (n=5714) mean±SD, median (range) or n (%) |
| --- | --- | --- |
| Age at marriage (years) | 20.1±3.2 | 28.2±6.1 |
| Maternal education | | |
| Literate | 1198 (33.1) | 1934 (38.4) |
| Women's employment status | | |
| Employed | 1129 (31.2) | – |
| Husband's employment status | | |
| Employed | 3582 (98.9) | – |
| Wealth Index (quintiles) | | |
| Q1 (least wealthy) | 791 (21.8) | – |
| Q2 | 659 (18.2) | |
| Q3 | 736 (20.3) | |
| Q4 | 813 (22.4) | |
| Q5 (wealthiest) | 624 (17.2) | |
| Reported smoking | 95 (2.6) | – |
| Current use of smokeless tobacco | 818 (21.9) | – |
| Total no of pregnancies | 3 (1–16) | 3 (1–20) |
| Primigravida | 836 (22.4) | 826 (16.3) |
| Multigravida (2–5) | 2232 (59.8) | 2576 (51) |
| Grand multigravida (>5) | 663 (17.8) | 1654 (32.7) |
| No of previous livebirths | 2 (1–14) | 3 (1–15) |
| No of previous stillbirths | 1 (1–5) | 1 (1–9) |
| No of previous miscarriages | 1 (1–10) | 1 (1–9) |
| No of previous induced abortions | 1 (1–3) | 1 (1–5) |

ANC, antenatal care; PRISMA, Pregnancy Risk Infant Surveillance and Measurement Alliance.

verbal autopsy is performed using the WHO 2016 verbal autopsy instrument[9] by trained research staff and coded by a physician to ascertain cause of death.

All visits are captured electronically through a custom-built Android application. This application uses an Open Smart Register Platform, an open-source mobile health platform designed to support the WHO SMART guidelines for data-driven decision-making in healthcare.[10] Once entered into the application, data are extracted, transformed and loaded into a PostgreSQL[11] relational database management system to allow for monitoring, reporting, and analytics. The application is user-specific and team-based (midwives, CHWs), with each team having a unique ID and password for data collection. The coordinators at the study sites perform period data checks and monitoring in order to ensure accurate data collection.

### Patient and public involvement

Written informed consent is obtained from all participants in the local language (Urdu). For illiterate participants, consent was obtained through a trusted witness who explained the consent form in the participant's language.

A thumbprint or mark was taken from the participant, with both the participant and the witness signing the form to validate the process. The cohort participants are not involved in the design or recruitment of the study. However, the community mobilisation teams of PRISMA Pakistan have been actively engaged with community stakeholders and pregnant women to inform on the study and benefits of early initiation of ANC.

### Strengths and limitations of this study

The prospective nature of the PRISMA Pakistan cohort provides pregnancy outcomes from the attendees of the PHC available in the community within the specified geographical area in a periurban community in Pakistan. This cohort leverages an existing demographic surveillance system to capture pregnancies and follow pregnant women and neonates with low refusal and attrition rates (<10%) as compared with other similar cohorts in the literature.[4 5] The PRISMA study would include 4–5 antenatal touchpoints, labour and delivery information followed by six postnatal touchpoints till 1 year after delivery. These interactions with well-constructed

**Table 2** Pregnancy characteristics of enrolled participants

| Characteristics | Total population (n=3731) Mean±SD; n (%) |
|---|---|
| Age at the time of pregnancy (years) | 26.6±5.8 |
| No of ANC visits | 5.4±2.8 |
| GA at first ANC visit (weeks) | 17.2±8.1 |
| MUAC (cm) | 25.8±4.1 |
| Nutritional status based on MUAC (cm) | |
| Normal (≥23) | 2724 (73.0) |
| Malnourished (<23) | 1006 (27.0) |
| BMI (kg/m$^2$) | 23.4±5.2 |
| Nutritional status based on BMI (kg/m$^2$) | |
| Underweight (<18.5) | 564 (15.1) |
| Normal weight (18.5–22.9) | 1454 (39.0) |
| Overweight (23.0 to <26.9) | 891 (23.9) |
| Obese (≥27.0) | 821 (22.0) |
| Haemoglobin levels (g/L) | 103±14 |
| Anaemia status (haemoglobin level) | |
| Normal (≥110 g/L) | 1161 (35.7) |
| Mild anaemia (100–109 g/L) | 940 (28.9) |
| Moderate anaemia (70–99 g/L) | 1077 (33.1) |
| Severe anaemia (<70 g/L) | 71 (2.2) |
| Chronic hypertension | 124 (3.3) |
| Diabetes mellitus | 23 (0.6) |
| Gestational hypertension | 48 (1.3) |
| Hepatitis B | 33 (1.2) |
| Hepatitis C | 52 (1.9) |

ANC, antenatal care; BMI, body mass index; GA, gestational age; MUAC, mid-upper arm circumference.

**Table 3** Delivery characteristics and neonatal outcomes in the study population

| Characteristics | n (%) |
|---|---|
| No of pregnant women | 3731 |
| No of fetuses | |
| Single | 3681 (98.7) |
| Multiple gestation | 50 (1.3) |
| Place of delivery* | |
| Partner referral facility | 2288 (64.0) |
| Other facilities | 502 (14.0) |
| Home births | 786 (22.0) |
| Mode of delivery* | |
| Vaginal | 2620 (73.3) |
| Spontaneous | 2617 (99.9) |
| Assisted | 3 (0.1) |
| Caesarean | 956 (26.7) |
| Elective | 530 (55.7) |
| Emergent | 422 (44.3) |
| No of birth outcomes | 3781 |
| Birth outcomes† | |
| Livebirths | 3478 (92.0) |
| Stillbirths | 98 (2.6) |
| Miscarriages | 180 (4.8) |
| Abortions | 25 (0.7) |
| Gender* | |
| Male | 1836 (51.3) |
| Female | 1738 (48.6) |
| Ambiguous | 2 (0.0) |
| Gestational age at delivery (weeks)‡ | |
| Extremely preterm (<28) | 8 (0.2) |
| Very preterm (28 to <32) | 44 (1.3) |
| Moderate-to-late preterm (32 to <37) | 644 (18.6) |
| Term (≥37) | 2761 (79.9) |
| Birthweight categories (kg)‡§ | |
| Extremely low birth weight (<1) | 2 (0.1) |
| Very low birth weight (1 to < 1.5) | 31 (1.2) |
| Low birth weight (1.5 to <2.5) | 578 (22.8) |
| Normal birth weight (≥2.5) | 1930 (75.9) |

*Reported for livebirths and stillbirths only.
†n=50 women had multiple gestations.
‡Reported for livebirths only; missing ultrasound on n=21, hence GA at delivery could not be ascertained.
§Only reported for those collected within 72 hours; missing data on n=937 infants.
GA, gestational age.

variables at each time point would add considerably rich information regarding pregnancy outcomes in this population. For the current description, we have reported birth and neonatal indicators only, but the main PRISMA study will have infant morbidity and mortality outcomes until 1 year of life. However, there are certain limitations to this work. Similar to the Global Network Maternal Newborn Health Registry, in our current work, women seek ANC in the second trimester[5]; therefore, our estimates regarding early-pregnancy outcomes, such as miscarriage/abortion, are underestimated given the mean GA at enrolment of 17 weeks. However, for the next phase of the PRISMA study, community mobilisation efforts will be put in place so that majority of the women enrol in the first trimester of pregnancy. Information on some key variables, such as pregnancy history as well as labour and delivery outcomes for those women who delivered at home and outside the recommended referral facility, is self-reported, thus limiting its interpretation. Another limitation is that not all women from the community are seeking care during pregnancy, and hence, may be different from those enrolled in PRISMA Pakistan, thus limiting the generalisability. This phenomenon has also been reported in other studies where women who do not seek ANC services or

seek services later in their pregnancy may be more disadvantaged in terms of their sociodemographic characteristics,[12 13] thus predisposing them to poorer outcomes.[14] However, with the existing baseline data acquired

**Table 4** Rates of maternal and neonatal mortality and stillbirth in PRISMA Pakistan

| Indicators | Mortality rates with (95% CI) in the study cohort | Mortality rates in Pakistan (2018–19) |
| --- | --- | --- |
| Maternal mortality rate | 172 (63 to 375)/100 000 livebirths | 186/100 000 livebirths |
| Stillbirth rate | 27 (22 to 33)/1000 births | 44/1000 births |
| Neonatal mortality rate | 52 (45 to 60)/1000 livebirths | 51/1000 livebirths |

PRISMA, Pregnancy Risk, Infant Surveillance and Measurement Alliance.

through the demographic surveillance system, the study team will be able to compare any differences in health-seeking behaviours of women who did not participate in the study.

## Findings to date

The sociodemographic and clinical characteristics of women included, as well as those excluded from the study, are described in table 1.

As of August 2022, 3731 women with pregnancy outcomes serve as a baseline for the PRISMA Pakistan cohort. Women enrolled in this cohort have a mean age of 26.6±5.8 years at the time of pregnancy. The average number of antenatal visits were 5.4±2.8, with a mean GA of 17.2±8.1 weeks at the first visit. Using the cutoff of mid-upper arm circumference of <23 cm, 27% (n=1006) of pregnant were malnourished. Approximately two-thirds of women (64%, n=2570) had either mild, moderate or severe anaemia. Detailed pregnancy characteristics of these women are described in table 2.

Majority of the pregnancy outcomes in this cohort were live births (92%, n=3,478) followed by miscarriages/abortions (5.5%, n=205) and stillbirths (2.6%, n=98). Twenty-two percent (n=786) of these women delivered at home. Overall, 20.1% (n=696) infants were preterm, and 24.1% (n=611) were low birth weight (table 3).

There were 98 stillbirths among this cohort, with 75% intrapartum stillbirths (n=74) and 25% antepartum stillbirths (n=24). Nearly three-quarters of neonatal deaths (76.4%, n=139) occurred during the first week of life. The main cause of death during the neonatal period was perinatal asphyxia (39.6%, n=72), preterm birth (19.8%, n=36), infections (12.6%, n=23), and congenital malformations (4.9%, n=9). The remaining causes included other perinatal complications (5.5%, n=10), while for 32 cases (17.6%), the cause could not be ascertained.

In this cohort, the maternal mortality rate is 172 per 100 000 live births, stillbirth rate is 27 per 1000 births, while neonatal mortality rates in the PRISMA Pakistan cohort are 52 per 1000 live births. The mortality rates for PRISMA study and national mortality rates [15][16] have been described in table 4. However, since the current study may not have sufficient numbers to comment on maternal mortality, statistical comparison of these estimates was not performed.

**Acknowledgements** The authors acknowledge the contributions of all women who participated in the study.

**Collaborators** The study protocol for the next phase of this study, a harmonised multisite PRISMA open cohort is registered with https://clinicaltrials.gov (number: NCT05904145) or by contacting the corresponding author of this manuscript.

**Contributors** ZHoodbhoy, FJ, MIN, FF and ES conceptualised the study and secure funding. ZHoodbhoy, NY, and UK supervised the study at field sites. ZHoodbhoy, SN, AJ, NY, MK, ZHussain, and UK were involved in the analysis, interpretation, and writing manuscript. ZHoodbhoy, FJ, MIN, FF and ES reviewed and provided scientific revision to the manuscript. All authors agreed prior to submission to take responsibility and be accountable for the contents of the manuscript. All authors read and approved the final manuscript. ZHoodbhoy is responsible for the overall content as guarantor.

**Funding** This study is funded by the Bill & Melinda Gates Foundation (INV-005776).

**Map disclaimer** The inclusion of any map (including the depiction of any boundaries therein), or of any geographic or locational reference, does not imply the expression of any opinion whatsoever on the part of BMJ concerning the legal status of any country, territory, jurisdiction or area or of its authorities. Any such expression remains solely that of the relevant source and is not endorsed by BMJ. Maps are provided without any warranty of any kind, either express or implied.

**Competing interests** None declared.

**Patient and public involvement** Patients and/or the public were not involved in the design, or conduct, or reporting, or dissemination plans of this research.

**Patient consent for publication** Consent obtained directly from patient(s)

**Ethics approval** This study involves human participants and the PRISMA MNH Pakistan baseline study received ethical approval in Pakistan (001-VPT-IRB-20). Participants gave informed consent to participate in the study before taking part.

**Provenance and peer review** Not commissioned; externally peer reviewed.

**Data availability statement** Data are available on reasonable request.

**ORCID iDs**
Sabahat Naz http://orcid.org/0000-0003-4448-0061
Fyezah Jehan http://orcid.org/0000-0002-5874-4358
Zahra Hoodbhoy http://orcid.org/0000-0002-0439-8293

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
