## [Reviewer comments · BMJ Open]

ARTICLE DETAILS

TITLE (PROVISIONAL)	Cohort profile: The Pregnancy Risk Infant Surveillance and Measurement Alliance (PRISMA) - Pakistan
AUTHORS	Naz, Sabahat; Jaffar, Ali; Yazdani, Nida; Kashif, Muhammad; Hussain, Zaid; Khan, Uzma; Farooq, Fouzia; Nisar, Muhammad; Jehan, Fyezah; Smith, Emily; Hoodbhoy, Zahra

VERSION 1 – REVIEW

REVIEWER	Carl Bose University of North Carolina at Chapel Hill
REVIEW RETURNED	20-Aug-2023

GENERAL COMMENTS	General comments: This manuscript describes the Pregnancy Risk Infant Surveillance and Measurement Alliance (PRISM), a multi-country effort to collect granular data describing pregnancy risks and outcomes across five countries. This project has great promise. This manuscript is really a methods paper with some outcome data for sites in Pakistan. However, there are some significant problems in the presentation of the methods and inferences drawn from the initial data set. In the second paragraph of the Introduction, they justify the investment in this project by claiming that “there is a dearth of such longitudinal data”. This statement dismisses the fact that there are indeed other sources of these data. They would strengthen their argument by contrasting the PRISM project with other longstanding projects. Two come to mind: the DHS program and the Maternal Newborn Health Registry of the NICHD Global Network for Women's and Children's Health Research. The latter uses very similar methodology and has generated a wealth of data over more than a decade. How is the PRISM project different from these longstanding datasets? What does it add? As a methods paper, this manuscript leaves many details out. These must be included if the reader is to understand the value of these data (see below). In addition, there are significant problems with the preliminary data analyses, most notably the relatively small sample upon which to draw inferences. (See specific examples below.) Specific comments Methods 1. The study populations reside in two peri-urban communities in Karachi. To support the claim that they have captured a “population-base”, they make the statement that “existing maternal and child health surveillance system” is being leveraged. What is this system? How does the number of pregnancies reported in the PRISM database compare to those in this alternate system? How many
---

	women were excluded because of expected delivery outside of the catchment area? This latter information should be included in the Figure. 2. How was gestational age established? This is critical information for some outcomes. 3. How reliable is “pregnancy history”? 4. The mean gestational age at enrollment was 17.2 weeks. This makes all data regarding miscarriage and abortion suspect. More than ½ of women would be enrolled after these events occur. 5. Among women who delivered at home, how were birth weight data collected? At what postnatal age? 6. The percentage of women with gestational hypertension is surprisingly low (1.3%). Was blood pressure routinely measured? What method was used? 7. Were tests for hepatitis B and C routinely performed? 8. The mode of delivery does not include instrumented vaginal delivery. Were vacuum extraction and forceps never used? 9. How was cause of death assigned? 10. Historically, in many resource-limited settings, early neonatal death is often misclassified as a fresh stillbirth. Do the authors have any sense about the magnitude of this problem? Were any strategies used to minimize the potential impact of misclassification? 11. There is no documentation of ethics approval. Was informed consent of subjects required? If yes, how was it obtained? 12. Was the trial registered? Results 1. Mortality rates should be reported with confidence intervals. This addition will help readers interpret comparisons, e.g., to national statistics. Because of the relatively small sample, maternal mortality rates will have wide confidence intervals. Therefore, to conclude that mortality in this population is “lower than the national maternal mortality” (p. 11; line 50) is very suspect. If this statement had been subjected to statistical analysis, it is doubtful that the point estimate would have been statistically different than national estimates. 2. In the Strengths and Limitations section, the authors list as a key strength the ability to “capture pregnancies in the first trimester”. Their data does not support this claim, based on a mean gestational age of enrollment of 17.2 weeks. In fact, I would consider this a limitation. 3. There is no data to support their claim that the study population is “nationally representative”. They would need to provide evidence that two peri-urban areas in Karachi are nationally representative. 4. I do not understand the statement that “nearly 40% may deliver outside the health facility”. Is this a national statistic? Their data determined that 22% were home deliveries. Other In the bulleted list of strengths and limitations that accompany the abstract, bullet #2 describes this cohort as representative of “such settings in low-resource countries”. Pakistan has among the worst pregnancy outcomes in the world. Data from Pakistan would generally be considered representative of few settings in other LMICs.
--	--

REVIEWER	James Berkley KEMRI/Wellcome Trust Research Programme, Clinical Research
REVIEW RETURNED	22-Aug-2023

GENERAL COMMENTS	This is a vital area of need for better data to inform maternal and
---

	newborn health programmes and targeted interventions. The manuscript is clearly written, however there are several areas where more details and precision in descriptions and findings, more depth in the discussion with key references are needed. The cohort description section needs to be expanded. More details are needed on the existing pregnancy surveillance system as they can vary in quality, coverage and completeness. Is the surveillance run by the government? Is it through facilities or community health workers etc? Does it cover all women or are some not included in certain geographic areas or because they are attending private healthcare? Are there any audit or performance figures for the surveillance? What is the average gestational age and what proportion are <20 weeks on detection by the system? Please include whether written informed consent was used and procedures for illiterate women. Please state if all women identified by the pregnancy surveillance system receive an ultrasound scan for gestational age assessment. Please include the number were excluded - details for the exclusions listed in the text, ectopic or molar pregnancies or those who plan to move outside the study catchment area during pregnancy or 12 months after delivery, are not presented in Figure 1. Also needed in Figure 1 and the text is the number lost to follow up before delivery (e.g. reached term + 2 weeks without a birth outcome recorded) and in the post-natal period. In the methods details of procedures to guard against loss to follow up should be documented. Depending on these results, in the Discussion, a comment on whether loss to follow up might have resulted in bias, such as missed miscarriage, stillbirth or early neonatal deaths, may have occurred as these are sometimes reasons for participants to feel uncomfortable about reporting within a study. It would be helpful to readers to include the online CRF and data dictionary as appendices or as URLs where they can be accessed. Were they based on any existing materials that can be referenced? For home births, what were the procedures for recording pregnancy outcome, birthweight etc, and how long after birth were these recorded? The fact that more than half of pregnant women were not enrolled because they did not visit a PHC may represent major bias. Can data be presented on ethnic groups and/or geographic or socioeconomic differences between the 5,714 not attending PHC and the 5,424 enrolled? This is important because the most at risk populations can be the hardest to reach when working through routine services. In the Discussion, potential enrolment and follow up biases should be discussed. In a study like this, bias is very hard to avoid and it is best that it is explored and reported. On page 11, could bias in enrolment account for lower than national average maternal mortality and stillbirth rates, or are these areas better served by healthcare and have a more affluent population than the national average? Would outcomes be expected to be different among women not attending PHC or making their first ANC visit after 20 weeks gestation? These should be discussed in relation to relevant literature, for example:
--	---

	https://academic.oup.com/eurpub/article/25/suppl_3/ckv176.078/2578903 https://bmcpregnancychildbirth.biomedcentral.com/articles/10.1186/s12884-016-0829-8 https://bmcpregnancychildbirth.biomedcentral.com/articles/10.1186/s12884-016-0979-8 https://journals.plos.org/plosmedicine/article?id=10.1371/journal.pmed.1004022 https://www.thelancet.com/journals/lancet/article/PIIS0140-6736(16)32254-1/fulltext https://pophealthmetrics.biomedcentral.com/articles/10.1186/s12963-023-00309-7 https://bmcpublichealth.biomedcentral.com/articles/10.1186/s12889-020-08480-4
--	--

VERSION 1 – AUTHOR RESPONSE

Reviewer: 1; Dr. Carl Bose, University of North Carolina at Chapel Hill

S.no	Reviewer's Comments	Responses
1.	In the second paragraph of the Introduction, they justify the investment in this project by claiming that "there is a dearth of such longitudinal data." This statement dismisses the fact that there are indeed other sources of these data. They would strengthen their argument by contrasting the PRISM project with other longstanding projects. Two come to mind: the DHS program and the Maternal Newborn Health Registry of the NICHD Global Network for Women's and Children's Health Research. The latter uses very similar methodology and has generated a wealth of data over more than a decade. How is the PRISM project different from these longstanding datasets? What does it add?	Thank you for your comments. We recognize the importance of providing a comprehensive explanation for the significance of our study. Indeed, there are existing sources of data, such as the DHS and the Maternal Newborn Health Registry of the NICHD Global Network for Women's and Children's Health Research, which provide valuable information that contributed significantly to the understanding of maternal and child health. However, DHS is a cross-sectional survey and with very few clinical measurements. The NICHD Global Network surveillance programs collect limited data about the underlying surveillance population; clinical data is collected only for women who enrol in the nested clinical trials and this clinical data is largely related to the primary and secondary outcomes of the trials. However, the PRISMA cohort aims to collect very detailed clinical information throughout the pregnancy, i.e., at enrolment, at antenatal care visits 20, 28, 32, 36, at delivery and up to one year postnatal. This has been added on page 5, Lines 82-89 of the manuscript.
2.	The study populations reside in two peri-urban communities in Karachi. To support the claim that they have captured a "population-base", they make the statement that "existing maternal and child health surveillance system" is being leveraged. What is this system? How does the number of	The Department of Pediatrics and Child Health at Aga Khan University has been conducting maternal and child surveillance at the peri-urban coastal field sites since 2010. This system gathers data on vital events regarding married women of reproductive age and children under 5 years old. As depicted in the figure, at the baseline, approximately 50% of women were availing antenatal care services at the PHC, while the

	pregnancies reported in the PRISM database compare to those in this alternate system? How many women were excluded because of expected delivery outside of the catchment area? This latter information should be included in the figure.	remaining half either sought care at different facilities or opted for no care at all. This has been added on page 6, Lines 106-108 of the manuscript.
3.	How was gestational age established? This is critical information for some outcomes.	The Gestational age was established via Ultrasound (GE Vivid IQ) at the time of enrolment. This has been added on page 6, Line 111 of the manuscript.
4.	How reliable is "pregnancy history"?	Pregnancy history relies solely on self-reported data thus limiting its interpretation in our setting.
5.	The mean gestational age at enrollment was 17.2 weeks. This makes all data regarding miscarriage and abortion suspect. More than ½ of women would be enrolled after these events occur.	We have noted this issue in the limitations and noted that it should not be interpreted as a prevalence estimate.
6.	Among women who delivered at home, how were birth weight data collected? At what postnatal age?	This is an important question. The surveillance and mobilization teams notify births that occur at home or at partner facilities. The CHWs then try to visit the family within 72 hours to capture relevant information. This has been added on pages 6-7, Lines 116-126 of the manuscript.
7.	The percentage of women with gestational hypertension is surprisingly low (1.3%). Was blood pressure routinely measured? What method was used?	Blood pressure is routinely measured at all ANC visits using the MICROLIFE AG 9443. For the purpose of this analysis, hypertension was defined as any two readings that were ≥ 140 systolic BP or ≥ 90 diastolic BP during these ANC visits. We understand that this is a crude method of estimating hypertension. However, we believe that once the PRISMA Pakistan cohort is established, we will have more accurate data on gestational hypertension. For perspective, the Community level intervention for preeclampsia (CLIP) trial reported that the incidence of gestational hypertension in Pakistan's cohort was around 3% https://doi.org/10.1371/journal.pmed.1002783.

8.	Were tests for hepatitis B and C routinely performed?	During the baseline data collection, all women visiting the PHC underwent standard tests, which included screening for hepatitis B and C. This has been added on page 6, Line 120 of the manuscript.
9.	The mode of delivery does not include instrumented vaginal delivery. Were vacuum extraction and forceps never used?	Among the 2620 women who had a normal vaginal delivery, only 3 (0.1%) required the use of forceps. This information has been added in Table 3 on page 10 of the manuscript.
10.	How was cause of death assigned?	In case of either maternal or infant mortality, a verbal autopsy is performed using the WHO 2016 verbal autopsy instrument by trained research staff and coded by a physician to ascertain cause of death. This has been added on pages 6-7, Lines 124-126 of the manuscript.
11.	Historically, in many resource-limited settings, early neonatal death is often misclassified as a fresh stillbirth. Do the authors have any sense about the magnitude of this problem? Were any strategies used to minimize the potential impact of misclassification?	Misclassification of neonatal death and stillbirths is problematic in resource limited settings. However, the verbal autopsy team that is employed in these areas have been extensively trained by WHO master trainers and have several years of experience thus minimizing the risk of misclassification including collection information about whether the newborn cried or showed other signs of life after delivery.
12.	There is no documentation of ethics approval. Was informed consent of subjects required? If yes, how was it obtained?	We have added the information in the ethical approval section of the manuscript. This has been added on page 7, Lines 141 of the manuscript.
13.	Was the trial registered?	The study protocol for the next phase of this study, a harmonized multi-site PRISMA open cohort is registered with https://clinicaltrials.gov (number: NCT05904145) The same registration number has been added in the abstract as Trial registration number NCT05904145 on Page 3, Line 53 as well as on Page 12, Lines 197-198.
14.	Mortality rates should be reported with confidence intervals. This addition will help readers interpret comparisons, e.g., to national statistics. Because of the relatively small	Thank you for the comment. We understand the importance of reporting mortality rates with confidence intervals, particularly for facilitating comparisons with national statistics. We reported the confidence intervals for the mortality rates in our study to provide a measure of precision.

	sample, maternal mortality rates will have wide confidence intervals. Therefore, to conclude that mortality in this population is "lower than the national maternal mortality" (p. 11; line 50) is very suspect. If this statement had been subjected to statistical analysis, it is doubtful that the point estimate would have been statistically different than national estimates.	This information has been added in Table 4 on page 11 of the manuscript.
15.	In the Strengths and Limitations section, the authors list as a key strength the ability to "capture pregnancies in the first trimester". Their data does not support this claim, based on a mean gestational age of enrollment of 17.2 weeks. In fact, I would consider this a limitation.	Thank you for the observation. This has been corrected and been stated on page 11, Lines 184-187 of the manuscript.
16.	There is no data to support their claim that the study population is "nationally representative". They would need to provide evidence that two peri-urban areas in Karachi are nationally representative.	We have updated the information in the strengths and limitations section of the manuscript to reflect that the population included in this cohort is likely to be representative of individuals from low-resource settings. This has been added on page 4, Lines 69-70 as well as on Page 11, Lines 181-182 of the manuscript.
17.	I do not understand the statement that "nearly 40% may deliver outside the health facility". Is this a national statistic? Their data determined that 22% were home deliveries.	To clarify, the 36% comprises of both the 22% home births and an additional 14% of women who delivered outside of the recommended referral facility. This information has been clarified in table 3 on page 10 as well as in the bulleted list of strengths and limitations on Page 4, Lines 71-73 of the manuscript.
18.	In the bulleted list of strengths and limitations that accompany the abstract, bullet #2 describes this cohort as representative of "such settings in low-resource countries". Pakistan has among the worst pregnancy outcomes	Thank you for the comment. We have revised bullet # 2 to reflect its generalizability on Page 4, Lines 69-70.

	in the world. Data from Pakistan would generally be considered representative of few settings in other LMICs.	
--	---	--

Reviewer 2: Dr. James Berkley, KEMRI/Wellcome Trust Research Programme

S.No	Reviewer's comments	Responses
1.	The cohort description section needs to be expanded. More details are needed on the existing pregnancy surveillance system as they can vary in quality, coverage and completeness. Is the surveillance run by the government? Is it through facilities or community health workers etc? Does it cover all women or are some not included in certain geographic areas or because they are attending private healthcare? Are there any audit or performance figures for the surveillance? What is the average gestational age and what proportion are <20 weeks on detection by the system?	Thank you for your comment. This information has been added on pages 6-7, Lines 106-132 of the manuscript.
2.	Please include whether written informed consent was used and procedures for illiterate women	Thank you for the comment. The consent procedure for literate and illiterate participants has been described on page 7, Lines 134-136 of the manuscript.
3.	Please state if all women identified by the pregnancy surveillance system receive an ultrasound scan for gestational age assessment.	Women residing in the catchment area can visit the PHC to access antenatal care. All women who opt to visit PHC undergo an ultrasound scan for GA assessment. This is currently only 50% of the existing surveillance catchment as shown in figure 1, submitted as a separate PDF file.
4.	Please include the number were excluded – details for the exclusions listed in the text, ectopic or molar pregnancies or those who plan to move outside the study catchment area during pregnancy or 12 months after delivery, are not presented in Figure 1.	Thank you for raising this. The exclusion criteria will apply to women who enroll in the PRISMA Pakistan cohort. The current study only reflects the baseline profile of women seeking care at the PHC and whose outcomes are available. This information has been added in Figure 1, submitted as a separate PDF file.
5.	Also needed in Figure 1 and the text is the number lost to follow-up before	Loss to follow up is a very important indicator to track. For this analysis, the loss to follow up rate was 5%. This

	delivery (e.g. reached term + 2 weeks without a birth outcome recorded) and in the post-natal period. In the methods, details of procedures to guard against loss to follow up should be documented. Depending on these results, in the Discussion, a comment on whether loss to follow up might have resulted in bias, such as missed miscarriage, stillbirth or early neonatal deaths, may have occurred as these are sometimes reasons for participants to feel uncomfortable about reporting within a study.	information has been added in Figure 1, submitted as a separate PDF file.
6.	It would be helpful to readers to include the online CRF and data dictionary as appendices or as URLs where they can be accessed. Were they based on any existing materials that can be referenced?	We agree with the reviewer and plan to publish the harmonized, multi-site PRISMA study CRFs with the study protocol paper.
7.	For home births, what were the procedures for recording pregnancy outcome, birthweight etc, and how long after birth were these recorded?	The details regarding home deliveries and the information capture have been added on page 6, Lines 122-124 of the manuscript.
8.	The fact that more than half of pregnant women were not enrolled because they did visit a PHC may represent major bias. Can data be presented on ethnic groups and/or geographic or socioeconomic differences between the 5,714 not attending PHC and the 5,424 enrolled? This is important because the most at risk populations can be the hardest to reach when working through routine services. In the Discussion, potential enrolment and follow up biases should be discussed. In a study like this, bias is very hard to avoid and it is best that it is explored and reported. On page 11, could bias in enrolment account for lower than national average maternal mortality and stillbirth rates, or are these areas better served by healthcare and have a more affluent population than the national average? Would outcomes be expected to be different among	We agree that the health seeking behavior of those who did not seek ANC services maybe significantly different. We have added the available sociodemographic and clinic information of these women in table 1 on page 8 of the manuscript. We anticipate that due to the extensive community mobilization planned for the PRISMA study, the number of women not seeking ANC services would reduce. Despite this, there would be a considerable population that do not seek these services. However, due to surveillance activities we will be able to capture basic sociodemographic and pregnancy outcome information on them. This will be presented in the PRISMA study analysis as the insights from these may have implications on the results and eventual implementation of programs. This has been discussed in light of the references provided on page 11, Lines 189-193 of the manuscript.

	women not attending PHC or making their first ANC visit after 20 weeks gestation? These should be discussed in relation to relevant literature, for example: https://academic.oup.com/eurpub/article/25/suppl_3/ckv176.078/2578903 https://bmcpregnancychildbirth.biomedcentral.com/articles/10.1186/s12884-016-0829-8 https://bmcpregnancychildbirth.biomedcentral.com/articles/10.1186/s12884-016-0979-8 https://journals.plos.org/plosmedicine/article?id=10.1371/journal.pmed.1004022 https://www.thelancet.com/journals/lancet/article/PIIS0140-6736(16)32254-1/fulltext https://pophealthmetrics.biomedcentral.com/articles/10.1186/s12963-023-00309-7 https://bmcpublichealth.biomedcentral.com/articles/10.1186/s12889-020-08480-4	
--	---	--

VERSION 2 – REVIEW

REVIEWER	Carl Bose University of North Carolina at Chapel Hill
REVIEW RETURNED	25-Oct-2023

GENERAL COMMENTS	General comments This is my second review of this manuscript; significant revisions have improved the quality of the manuscript since my first review. For example, the authors have added crucial details about methodology. However, significant problems remain. My major concern is the derivation of the cohort and the conclusions by the authors that this cohort is “a good representation of those residing in similar low resource settings”. This may be so; the authors do provide demographic data that would allow a reader to draw inferences about the applicability of these data to other populations. However, it is not clear to me that this cohort represents a “typical” population in LMICs and even in Pakistan. They make no claims about this being a population-based cohort, which is the ideal study cohort. But within the study catchment area there appear to be important differences between the study cohort and the general population. For example, in Table 1, they compare the enrolled and unenrolled cohorts from pregnancies in the catchment area identified by their surveillance system. There appear to be potentially significant differences in these populations that might impact
--

pregnancy outcomes (e.g., maternal age and literacy). Differences between this cohort and other populations in Pakistan regarding significant aspects of maternal care appear likely. For example, the c-section rates among Pakistani women in the NICHD Global Network Maternal Newborn Health Registry was 18.8% in 2022 compared to 26% in the PRISMA cohort. And it still not clear to me how the surveillance system works and the extent to which it captures all pregnancies in the catchment area. The authors acknowledge most of these shortcoming in the section entitled Strengths and Limitations. I would have preferred a more in-depth discussion about how the limitations might have influenced results with comparisons to published data describing other cohorts in Pakistan.

They continue to make comparisons and inferences without statistical tests (Page 12, lines 9-23). Any references to “higher” and “lower” with subsequent explanations of potential cause should either be abandoned (my preference) or acknowledge the possibility that the differences could have occurred by chance alone. The most egregious offense in this regard is the comparison of maternal mortality in the cohort compared to national statistics. By my calculation, there were only six maternal deaths in the cohort during the study period. Given the unlikelihood of maternal deaths, witnessed by the large confidence intervals, the sample size of the cohort does not support conclusions.

One of the strengths of the PRISMA Study is that includes follow up to one year. For example, in the NICHD Global Network, their Maternal Newborn Health Registry truncates follow up at 42 days postpartum. The authors would strengthen their argument for the value added of the PRISMA data by reporting one-year infant outcomes, particularly if their rate of follow up was high.

Specific Comments

Page 5, line 29. The word “either” should be followed by “or” at some point. I suggest after “(i.e., DHS)”.

Page 6, line 6. They begin this section with the statement: “the PRISMA study will be conducted in two peri-urban community settings”. I do not understand the future tense. They report the results of data collection. Has the study not already begun?

Page 7, lines 22-39. As noted in my general comments, I do not understand how the existing maternal and child health surveillance system works and to what extent captures all pregnancies. I could seek information in reference #6, but a brief overview would be helpful to the reader.

Table 3. There appear to be differences in the numbers reported in the text and the Figure and Table 3. For example, the total number of pregnancies for which data were available from the Figure is 3731. The total number of the population reported in Table 3 is 3576. I could not generate this number by accounting for twin births and/or subtracting miscarriages/abortions. Thereafter, few of the totals within categories equaled 3576. I assume that this resulted from missing data points or a single data point for twin gestations. However, it was confusing and diminished my confidence in the data. An explanation of the numbers would help.

	Page 12, lines 3-6. The reported causes of death represented only 72% of deaths. A list of the other causes that constituted 28% of deaths (the second largest category) would be useful.
--	---

REVIEWER	James Berkley KEMRI/Wellcome Trust Research Programme, Clinical Research
REVIEW RETURNED	20-Oct-2023

GENERAL COMMENTS	Several of my comments have been addressed, however I still find it difficult to assess how representative this cohort is. The inclusion criteria are given as " Pregnant women aged 15 - 49 years with a confirmed intrauterine pregnancy of gestational age (GA) <20 weeks via standard ultrasound (GE Vivid IQ)" however it is not clear whether ultrasound is something offered to all women free of charge at both government and private clinics or something else. Which exact PCH clinics were sampled? What type of clinics are they? Are there other clinics in the same area that were not sampled? How were data collected from women who did not receive ANC services or did not consent?
--

VERSION 2 – AUTHOR RESPONSE

Reviewer: 1; Dr. Carl Bose, University of North Carolina at Chapel Hill

S.No	Reviewer's Comments	Responses
19	My major concern is the derivation of the cohort and the conclusions by the authors that this cohort is "a good representation of those residing in similar low resource settings". This may be so; the authors do provide demographic data that would allow a reader to draw inferences about the applicability of these data to other populations. However, it is not clear to me that this cohort represents a "typical" population in LMICs and even in Pakistan. They make no claims about this being a population-based cohort, which is the ideal study cohort. But within the study catchment area there appear to be important differences between the study cohort and the general population. For example, in Table 1, they compare the enrolled and unenrolled cohorts from pregnancies in the catchment area identified by their surveillance system. There appear to be potentially significant differences in these populations that might impact pregnancy outcomes (e.g., maternal age and literacy). Differences between this cohort and other populations in Pakistan regarding significant aspects of maternal care appear likely. For example, the c-section rates among Pakistani women in the NICHD Global Network Maternal Newborn Health Registry was 18.8% in 2022 compared to 26% in the	Thank you for your detailed comment. There are some differences between the PRISMA cohort and those who were not enrolled (as shown in Table 1). The main difference is that the included population is, on average, a bit younger than the general population in the study area; other characteristics such as the proportion literate are very similar. However, we may not be able to draw inferences for the broader Pakistani population. Considering this, we have rephrased our statement "a good representation of those residing in similar low resource settings" in bullet points lines 66-67 on page 4 as well as in the strengths and limitations in lines 207-208 on page 12 of the manuscript.

	PRISMA cohort.	
20	And it still not clear to me how the surveillance system works and the extent to which it captures all pregnancies in the catchment area. The authors acknowledge most of these shortcoming in the section entitled Strengths and Limitations. I would have preferred a more in-depth discussion about how the limitations might have influenced results with comparisons to published data describing other cohorts in Pakistan.	We have added more details regarding the surveillance activities in lines 110-112 on page 6 and lines 210 – 212 on page 12 of the manuscript. Due to our long-standing relationship with community stakeholders, the refusal rate for surveillance activities is approximately 5%. We have added more information regarding the baseline cohort and the potential cohort to be collected in the future in lines 212-218 on page 13 of the manuscript.
21	They continue to make comparisons and inferences without statistical tests (Page 12, lines 9-23). Any references to “higher” and “lower” with subsequent explanations of potential cause should either be abandoned (my preference) or acknowledge the possibility that the differences could have occurred by chance alone. The most egregious offense in this regard is the comparison of maternal mortality in the cohort compared to national statistics. By my calculation, there were only six maternal deaths in the cohort during the study period. Given the unlikelihood of maternal deaths, witnessed by the large confidence intervals, the sample size of the cohort does not support conclusions.	Thank you for this very important observation. We completely agree that any kind of comparison with national estimates may be inaccurate at this stage. Hence, we have modified the statements to ensure that such inferences and their explanations are removed. Please refer to lines 194-203 on page 12 of the manuscript.
22	One of the strengths of the PRISMA Study is that includes follow up to one year. For example, in the NICHD Global Network, their Maternal Newborn Health Registry truncates follow up at 42 days postpartum. The authors would strengthen their argument for the value added of the PRISMA data by reporting one-year infant outcomes, particularly if their rate of follow up was high.	Thank you for your comment. The baseline PRISMA study only followed pregnant women for 4 weeks after delivery. However, the new PRISMA study will be following the infants up to 1 year postnatally. We have specified this information in lines 212-217 on page 13 of the manuscript.
23	Page 5, line 29. The word “either” should be followed by “or” at some point. I suggest after “(i.e., DHS)”.	Thank you for your suggestion. We have added “or” right after (i.e., DHS) in line 89 on page 5 of the manuscript.
24	Page 6, line 6. They begin this section with the statement: “the PRISMA study will be conducted in two peri-urban community settings”. I do not understand the future tense. They report the results of data collection. Has the study not already begun?	Apologies for the mistake. We have corrected it as “is being” and removed “will be” in line 104 on page 6 of the manuscript.

25	Page 7, lines 22-39. As noted in my general comments, I do not understand how the existing maternal and child health surveillance system works and to what extent captures all pregnancies. I could seek information in reference #6, but a brief overview would be helpful to the reader.	Thank you for your comment. We have added a brief overview of the existing maternal and child health surveillance system in lines 110-112 on page 6 of the manuscript.
26	Table 3. There appear to be differences in the numbers reported in the text and the Figure and Table 3. For example, the total number of pregnancies for which data were available from the Figure is 3731. The total number of the population reported in Table 3 is 3576. I could not generate this number by accounting for twin births and/or subtracting miscarriages/abortions. Thereafter, few of the totals within categories equaled 3576. I assume that this resulted from missing data points or a single data point for twin gestations. However, it was confusing and diminished my confidence in the data. An explanation of the numbers would help.	Thank you for this comment. We understand that the previous table did cause some confusion. However, we have now clarified the number of each variable in the outcomes table and reported the number of missing values as well in the footnote. We hope that this will clarify. Please refer to Table 3 on page 10 of the manuscript.
27	Page 12, lines 3-6. The reported causes of death represented only 72% of deaths. A list of the other causes that constituted 28% of deaths (the second largest category) would be useful.	Thank you for your comment. The three main causes of neonatal deaths (NND) were perinatal asphyxia (39.6%, n=72), preterm birth (19.8%, n=36), and infections (12.6%, n=23), which represented 72% of total NND. The distribution of the remaining 28% of NND includes congenital malformation (4.9%, n=9), other perinatal complications (5.5%, n=10), and unknown (17.6%, n=32). We have added this information in lines 190-193 on page 12 of the manuscript.

Reviewer 2: Dr. James Berkley, KEMRI/Wellcome Trust Research Programme

S.No	Reviewer's comments	Responses
9.	Several of my comments have been addressed, however I still find it difficult to assess how representative this cohort is.	Thank you for your comment. In terms of representative nature of the cohort, we have rephrased our statements in bullet points in lines 66-67 on page 4 and lines 207-208 on page 12 of the manuscript.
10.	The inclusion criteria are given as " Pregnant women aged 15 - 49 years with a confirmed intrauterine pregnancy of gestational age (GA)	Thank you for your comment. Each study site has its own primary

	<20 weeks via standard ultrasound (GE Vivid IQ)" however it is not clear whether ultrasound is something offered to all women free of charge at both government and private clinics or something else.	health care (PHC) centre that has been established and operated by the Aga Khan University. These PHCs are accessible to populations within their catchment area and provide free antenatal care services to all women visiting the PHC, including ultrasound. This has been clarified in lines 114-116 on page 6 of the manuscript.
11.	Which exact PCH clinics were sampled? What type of clinics are they? Are there other clinics in the same area that were not sampled? How were data collected from women who did not receive ANC services or did not consent?	Each study site established its own PHC clinic that is operated by the Department of Pediatrics and Child Health, Aga Khan University. The PHC at each site is accessible by the women of reproductive age (15 to 49 years) within the catchment area to avail free antenatal care services, including the ultrasound and laboratory tests, intrapartum (delivery), and postnatal care services. This information has been added in lines 115-118 on page 6 of the manuscript. In addition, the Department of Pediatrics and Child Health, Aga Khan University also run Health and Demographic Surveillance system since 2003 in four peri-urban low socioeconomic communities of Karachi. and adult female. Details of pregnant women who did not receive ANC services or did not consent for the study were obtained from the surveillance system. This information has now been added in lines 110-116 on page 6 and lines 210 – 212 on page 12 of the manuscript

VERSION 3 – REVIEW

REVIEWER	James Berkley KEMRI/Wellcome Trust Research Programme, Clinical Research
REVIEW RETURNED	20-Nov-2023
GENERAL COMMENTS	This revised manuscript provides the essential details of the cohort, the epidemiological framework, relationship to the demographic surveillance and comparison with the excluded group. I am slightly concerned about use of the term 'population-based' as a description

	of the cohort as the majority of the population are not included and there are some biases in the included group. 'cohort of AKU clinic attendees' might be more suitable.
--	--

VERSION 3 – AUTHOR RESPONSE

Reviewer 2: Dr. James Berkley, KEMRI/Wellcome Trust Research Programme

S.No	Reviewer's comments	Responses
1.	I am slightly concerned about use of the term 'population-based' as a description of the cohort as the majority of the population are not included and there are some biases in the included group. 'cohort of AKU clinic attendees' might be more suitable.	Thank you for your comment. We agreed with your suggestion and have revised the term "population-based" to "attendees of the PHC available in the community" in the bullet points on Page 4, lines 66-70 as well as in the strength and limitation section on Page 11, lines 202-203.